# White Blood Cells and Severe COVID-19: A Mendelian Randomization Study

**DOI:** 10.3390/jpm11030195

**Published:** 2021-03-12

**Authors:** Yitang Sun, Jingqi Zhou, Kaixiong Ye

**Affiliations:** 1Department of Genetics, Franklin College of Arts and Sciences, University of Georgia, Athens, GA 30602, USA; Yitang.Sun@uga.edu (Y.S.); Jingqi.Zhou@uga.edu (J.Z.); 2School of Public Health, Shanghai Jiao Tong University School of Medicine, Shanghai 200025, China; 3Institute of Bioinformatics, University of Georgia, Athens, GA 30602, USA

**Keywords:** COVID-19, white blood cells, basophils, Mendelian randomization

## Abstract

Increasing evidence shows that white blood cells are associated with the risk of coronavirus disease 2019 (COVID-19), but the direction and causality of this association are not clear. To evaluate the causal associations between various white blood cell traits and the COVID-19 susceptibility and severity, we conducted two-sample bidirectional Mendelian Randomization (MR) analyses with summary statistics from the largest and most recent genome-wide association studies. Our MR results indicated causal protective effects of higher basophil count, basophil percentage of white blood cells, and myeloid white blood cell count on severe COVID-19, with odds ratios (OR) per standard deviation increment of 0.75 (95% CI: 0.60–0.95), 0.70 (95% CI: 0.54–0.92), and 0.85 (95% CI: 0.73–0.98), respectively. Neither COVID-19 severity nor susceptibility was associated with white blood cell traits in our reverse MR results. Genetically predicted high basophil count, basophil percentage of white blood cells, and myeloid white blood cell count are associated with a lower risk of developing severe COVID-19. Individuals with a lower genetic capacity for basophils are likely at risk, while enhancing the production of basophils may be an effective therapeutic strategy.

## 1. Introduction

Coronavirus disease 2019 (COVID-19) is caused by infection of a novel virus called Severe Acute Respiratory Syndrome Coronavirus 2 (SARS-CoV-2) that has rapidly spread, causing irreversible damage or even death [1,2,3,4]. A growing number of studies have found that immune cells may play important roles in the COVID-19 severity and susceptibility [5,6,7,8,9]. Since prior findings were based on observational studies, the causal associations between white blood cells (WBC) and COVID-19 remain unclear. Identifying host factors predisposing individuals to COVID-19 susceptibility and severity is urgently needed to improve primary prevention and develop treatment strategies.

Many observational studies have reported risk factors for COVID-19 susceptibility and severity. Besides age and gender, some pre-existing conditions are also well-known to be associated with an increased risk of severe COVID-19, such as cardiovascular disease, diabetes, chronic respiratory disease, hypertension, and cancers [4,10,11,12]. Furthermore, elevated WBC and neutrophil counts, and depleted lymphocyte count have been repeatedly observed in COVID-19 patients with severe outcomes, and the neutrophil-to-lymphocyte ratio has been proposed as prognostic biomarkers [1,5,13,14,15,16]. However, findings from recent observational studies are inconsistent, and it is difficult to infer the causal roles of WBC in severe COVID-19 due to confounding and reverse causation [17,18,19]. Most existing studies measured blood cell counts in patients with confirmed infection of SARS-CoV-2 and as a result, the hematological indices could have been modified by immune responses [20]. It is unknown if blood cell counts before infection are associated with the risk of developing severe COVID-19. Even if WBC are measured before infection, they are influenced by many exogenous and endogenous factors (e.g., age, gender, disease status, and medications), which will confound the observational associations [21,22]. To date, no existing research has been able to interrogate the causal role of WBC in COVID-19 severity and susceptibility.

Mendelian randomization (MR) utilizes genetic variants as instrumental variables to approximate the lifetime status of exposure and enable causal inference in observational data. The random allocation of alleles at conception and the natural directional effects of genetic variants on phenotypes empower MR estimates to be less plagued by confounding and reverse causality [23,24]. In this study, we used two-sample MR analyses to examine potential causal associations between various WBC traits and the risk of COVID-19 severity and susceptibility. Recent genome-wide association studies (GWAS) with thousands of cases and millions of controls have identified scores of genetic loci associated with the susceptibility and severity of COVID-19 [25]. Two recent GWAS on WBC traits enrolled approximately half a million European-ancestry participants and dramatically increased the number of instrumental variables [26,27,28]. Building on these largest-to-date GWAS for both the exposure and outcome, we conducted two-sample bidirectional MR analyses to disentangle the causal effects of various WBC traits on COVID-19 severity and susceptibility, or vice versa.

## 2. Materials and Methods

### 2.1. Study Design and Data Sources

A two-sample bidirectional MR study was conducted to examine the causal effects of various WBC traits on COVID-19, and vice versa (Figure 1). GWAS summary statistics of WBC traits were extracted from publicly available data (https://www.ebi.ac.uk/gwas/downloads/summary-statistics, accessed on 19 January 2021) [29]. Summary statistics were obtained from the most recent version of GWAS from the COVID-19 Host Genetics Initiative (https://www.covid19hg.org/ accessed on 20 January 2021) [25]. Publicly available data sources had previously received appropriate ethics and institutional review board approvals, and thus further ethical approvals were not required.

### 2.2. Instrumental Variables for WBC and COVID-19

In our primary discovery analysis, instrumental variables for WBC were selected based on two recent GWAS as independent genetic variants at the study-specific genome-wide significance (*p* < 5 × 10^−9^) [26,27]. The first is a GWAS meta-analysis of 11 WBC traits, which included 408,112 participants of European ancestry and was released by Vuckovic et al. [26]. The other is a GWAS of 6 WBC traits, which included 562,132 European-ancestry participants and was derived by Chen et al. [27]. In the replication analysis, we selected SNPs that were associated with 21 white blood cell traits at the study-specific genome-wide significance (*p* < 8.31 × 10^−9^) in a GWAS meta-analysis comprising 173,480 individuals of European ancestry. This GWAS is called the Astle et al. study [28].

The instrumental variables for COVID-19 were retrieved at the genome-wide significance (*p* < 5 × 10^−8^) from the largest GWAS meta-analysis of COVID-19 to date, by the COVID-19 Host Genetics Initiative (HGI, round 5, released on 18 January 2021) [25]. For COVID-19 severity, we used two GWAS with European-ancestry participants. The first, called study A2, compared patients confirmed as “very severe respiratory” COVID-19 (*N* = 5105) with the general population samples (*N* = 1,383,241). The other, called study B2, compared hospitalized COVID-19 patients (*N* = 9986) with the general population (*N* = 1,877,672). For COVID-19 susceptibility, summary statistics were obtained from a GWAS in participants of European ancestry, including patients with laboratory-confirmed COVID-19 infection or physician confirmed COVID-19 (*N* = 36,590) and the population controls (*N* = 1,668,938). This is called study C2.

To ensure independence among instrumental variables for each exposure, we applied linkage disequilibrium (LD) clumping based on r^2^ > 0.001 and discarded variants within 1-Mb distance from other SNPs with a stronger association. Effect alleles of genetic instruments were harmonized across the GWAS of the exposure and the outcome, and those not present in the GWAS of the outcome were removed. F statistics were calculated to assess instrument strength and *F* ≥ 10 indicates strong instruments [30].

### 2.3. Statistical Analysis

The causal association of each exposure with each outcome was evaluated using the inverse variance-weighted (IVW) method with a multiplicative random-effects model [24,31,32]. Horizontal pleiotropy occurs when SNPs exert a direct effect on the severe COVID-19 through pathways other than the hypothesized exposure. To evaluate the possible presence of horizontal pleiotropy, we calculated Cochran’s Q statistic for heterogeneity and conducted the intercept test associated with the MR-Egger method. Additional sensitivity analyses were performed with MR-Egger [24,32], weighted median (WM) [33], weighted mode methods [34], and Mendelian randomization pleiotropy residual sum and outlier (MR-PRESSO) test [35]. The MR-Egger estimates allow directional horizontal pleiotropic effects. The weighted median method provides robust causal estimates even when up to 50% of SNPs are invalid genetic instruments [33]. The weighted mode method requires that the largest number of instruments that identify the consistent causal effect derived from valid instruments even when the majority of instruments are invalid [34]. The MR-PRESSO test was utilized to correct for the presence of specific horizontal pleiotropic outlier variants via detected outlier removal [35]. The false discovery rate (FDR) approach was adopted to correct for multiple testing [36]. The nominal *p*-value < 0.05 but above the FDR significance threshold (*q*-value < 0.05) was considered as suggestive evidence. All MR analyses were conducted in R with the TwoSampleMR package (version 3.6.7) [37].

## 3. Results

### 3.1. Analysis Pipeline

We first aimed to investigate the causal effects of WBC traits on COVID-19 severity and susceptibility via the forward MR analysis. In our main analysis, we selected uncorrelated SNPs as instrumental variables for each WBC trait from two recent GWAS in 408,112 and 562,132 participants of European ancestry [26,27]. We then performed a replication analysis using genetic instruments from a previous GWAS meta-analysis that included 173,480 individuals of European ancestry [28]. On the other hand, to evaluate the causal effects of COVID-19 on WBC traits, we conducted a reverse MR analysis using COVID-19-associated SNPs as instrumental variables. All SNPs for the WBC traits (*F* statistic > 77.82) and COVID-19 (*F* statistic > 3458.84) are strong instruments. A schematic of the analysis pipeline is shown in Figure 1. Details about GWAS, from which we extracted summary-level data, are presented in Appendix A.

### 3.2. Forward MR Analysis of the Effects of WBC Traits on COVID-19

By applying a two-sample MR approach, we first investigated the causal associations of 12 WBC traits with COVID-19. A relatively large number of independent SNPs, ranging from 115 for basophil percentage of white blood cells to 469 for monocyte count, were selected as genetic instruments for each WBC trait (Appendix A).

Using very severe respiratory COVID-19 as the outcome (A2), we identified suggestive causal, negative associations of basophil count (OR = 0.75, CI: 0.60–0.95, *p* = 0.015), myeloid WBC count (OR = 0.85, CI: 0.73–0.98, *p* = 0.022), and basophil percentage of WBC (OR = 0.70, CI: 0.54–0.92, *p* = 0.011) based on the IVW MR estimates under a multiplicative random-effects model (Figure 2) (Appendix A). Causal estimates from MR-Egger, WM, and weighted mode methods revealed broadly concordant effect directions, although they are mostly not statistically significant, probably due to the reduced power of these two approaches [32]. MR-PRESSO analysis did not identify any outlier SNPs and yielded significant causal estimates for basophil count (*p* = 0.015), myeloid WBC count (*p* = 0.0022), and basophil percentage of WBC (*p* = 0.011). No evidence of heterogeneity in causal estimates was found by the Cochran Q statistic, and no evidence of horizontal pleiotropy was reported by the MR-Egger intercept test. In the replication analysis using genetic instruments from another GWAS meta-analysis of WBC traits (i.e., the Astle et al. study), we similarly observed the significant negative effects of basophil count and myeloid WBC count. Additionally, nominally significant negative associations were observed for total WBC count (*p* = 0.038), neutrophil count (*p* = 0.016), granulocyte count (*p* = 0.028), sum neutrophil eosinophil counts (*p* = 0.032), sum basophil neutrophil counts (*p* = 0.012) (Appendix A).

Using hospitalized COVID-19 as the outcome (B2), we also found negative associations of basophil count (OR = 0.83, CI: 0.71–0.97, *p* = 0.020), basophil percentage of WBC (OR = 0.78, CI: 0.65–0.93, *p* = 0.005), and myeloid WBC count (OR = 0.90, CI: 0.73–0.996, *p* = 0.041) (Figure 2) (Appendix A). The causal effect of basophil count (*q* = 0.014) and basophil percentage (*q* = 0.042) surpassed the FDR correction in this analysis. Moreover, genetically predicted lower total WBC count (OR = 0.89, CI: 0.80–0.98, *p* = 0.016) was associated with increased risk of hospitalized COVID-19 using IVW with the random-effect model (Figure 2) (Appendix A). MR-Egger, WM, weighted mode methods, and MR-PRESSO analysis revealed broadly consistent effect directions. No pleiotropy was identified in the MR-Egger test and a potential causal effect of WBC count was retained after removing outlier SNPs identified by MR-PRESSO analysis. In the replication analysis, basophil count, basophil percentage, and total WBC count were also negatively associated with hospitalized COVID-19 (Appendix A). Taken together these two outcomes of COVID-19 severity, very severe respiratory and hospitalized COVID-19 (A2 and B2), we demonstrated that basophil count, basophil percentage, myeloid WBC count, and total WBC count had consistent, negative effects on the risk of severe COVID-19.

We further investigated the causal associations of WBC traits with COVID-19 susceptibility (C2). None of the WBC traits was causally associated with COVID-19 susceptibility (Figure 2) (Appendix A). Similar findings were observed in the replication analysis (Appendix A). Overall, there was no evidence of associations between any WBC trait and COVID-19 susceptibility.

### 3.3. Reverse MR Analysis of the Effects of COVID-19 on WBC Traits

We further explored the causal effects of COVID-19 severity and susceptibility on each WBC trait in the reverse direction using COVID-19-associated SNPs as instrumental variables. We used nine SNPs (A2) and five SNPs (B2) for COVID-19 severity, and seven SNPs (C2) for COVID-19 susceptibility (Appendix A). The reverse MR analysis showed no significant association of genetically predicted COVID-19 severity or susceptibility with any WBC trait (Figure 3) (Appendix A). The null findings were confirmed in the replication analysis that used summary statistics from the Astle et al. study of WBC traits (Appendix A).

## 4. Discussion

To our knowledge, this is the first MR study evaluating the causal roles of WBC traits in COVID-19 severity and susceptibility. Overall, our results suggest potential causal protective effects of higher basophil count, basophil percentage of WBC, and myeloid WBC count on the severity of COVID-19. No evidence was found for causal effects of WBC traits on the susceptibility of COVID-19, or for reverse causation of COVID-19 on WBC traits.

Previous observational studies have frequently shown increased WBC count in the severe COVID-19 patients, when compared to healthy controls or mild COVID-19 patients [1,5,13,38]. However, observational studies may be influenced by confounding and cannot always distinguish symptoms from causes. Using genetically predicted WBC traits, our MR analysis showed that lower basophil count, basophil percentage of white blood cells, and myeloid white blood cell count may play causal roles in increasing the risk of severe COVID-19. COVID-19 postmortems have revealed that the loss of basophils from the blood is partially explained by its transfer into myelomonocytic lung infiltrates [7,39]. Basophils play important roles in modulating innate immune responses to infection and the subsequent tissue repair [40,41]. Individuals with low basophil count and basophil percentage may not present a strong enough innate immune response to the SARS-CoV-2 infection. Moreover, dysregulated activation of basophils also plays a role in regulating coagulation that may be pertinent to frequent thrombotic complications in severe cases of COVID-19 [42,43]. Immune system disorders have been suspected of playing roles in severe COVID-19 risk [44,45]. Basophils are known to potentiate humoral immune responses by producing interleukins 6 and 4 (IL-6 and IL-4) [46]. A recent immune-monitoring study of COVID-19 patients found that the relative abundance of basophils increases noticeably from acute to recovery phases and is positively associated with the level of anti-CoV-2 immunoglobulin G (IgG). These observations indicate that the extent of basophil depletion may affect the efficacy of IgG responses to the SARS-CoV-2 infection. The mediating factors between basophils and IgG responses may include IL-6, IL-4, or other unexplored factors, and the same study also observed a negative association between IL-6 and anti-CoV-2 IgG [47]. The complete elucidation of the potential mechanism warrants further investigation.

Previous systematic reviews and meta-analyses of observational studies have shown WBC traits to be associated with COVID-19 severity and susceptibility [5,6,7,9]. However, the associations between WBC traits and COVID-19 may be susceptible to reverse causation. Our reverse MR study did not find evidence for the effects of COVID-19 on any WBC trait. This result further supports the direct effect of basophils on severe COVID-19, ruling out the confounding of reverse causation. Therefore, further mechanistic understanding of basophil will shed light on the etiology of severe COVID-19 and provide multiple targets of intervention for prevention and treatment.

Lymphopenia, as a response to viral infection, has been frequently associated with severe COVID-19 risk [1,2,38,48]. Persistent eosinopenia after admission was associated with COVID-19 risk in previous retrospective and prospective observational studies [5,17,49,50,51]. However, these associations of WBC traits are more common in COVID-19 patients when compared to the reference range or healthy controls [17,18,19,25,50,51]. This discrepancy may reflect reverse causality in retrospective and prospective observational studies, with depleted lymphocyte and eosinophil counts as a result of the immune response to SARS-CoV-2 infection [52]. In our MR analysis, we found no evidence for the causal association between lymphocyte or eosinophils and the risk of COVID-19 severity and susceptibility. We showed that neutrophils are slightly associated with severe COVID-19, consistent with our recent MR results [53]. Our reverse MR analysis indicates no causal effects of severe COVID-19 on lymphocytes.

One assumption of MR is that the instrumental variable influences severe COVID-19 risk only through its effect on the specific WBC trait. The Cochran Q statistic did not reveal heterogeneity among our genetic instruments, and the MR-Egger intercept test also indicated no presence of pleiotropic effects for basophils. Pleiotropic instrumental variables, which are SNPs associated with other phenotypes directly or indirectly associated with the outcome, were detected by using the MR-Egger intercept and MR-PRESSO tests. Our MR study was performed with strong instrumental variables and adequate statistical power, and we have conducted extensive sensitivity analyses. The exposure and outcome GWAS were drawn from participants of European ancestry. Our MR results were replicated in at least one replication analysis with different GWAS of WBC to ensure robustness and reduce false positives. To assess whether WBC traits are associated with COVID-19 and to evaluate the direction of association, we applied a two-sample bidirectional MR study using the most up-to-date GWAS of WBC traits and COVID-19 severity and susceptibility.

Our study also has several limitations. Although we applied multiple sensitivity analyses, it is impossible to fully rule out that some genetic variants might be pleiotropic. Another limitation is that some GWAS of WBC traits and the HGI GWAS of COVID-19 have overlapped samples, which may bias the causal effect. F statistics of instrumental variables were estimated to mitigate this issue, and strong instruments are less susceptible to this bias. Although not all significant results remained after correction for multiple testing, we also utilized other GWAS of WBC and COVID-19 to replicate the MR results. The replicated observations across multiple analyses reduce the chance of false positives. Lastly, we emphasize that our results should be interpreted with caution, and future studies are needed to elucidate the mechanical roles of WBC traits in severe COVID-19.

## 5. Conclusions

Our MR results suggest that there are inverse causal associations between basophil count, basophil percentage, and myeloid WBC count and the severity of COVID-19. No causal effects on the susceptibility of COVID-19 were found. There is no evidence of reverse causation of COVID-19 severity and susceptibility on WBC traits. Overall, this study provides valuable insights that individuals with a lower genetic capacity for basophils are likely at risk of the severe forms of COVID-19, and that enhancing the production of basophils may be a potential prevention and treatment strategy.

## Figures and Tables

**Figure 1 jpm-11-00195-f001:**
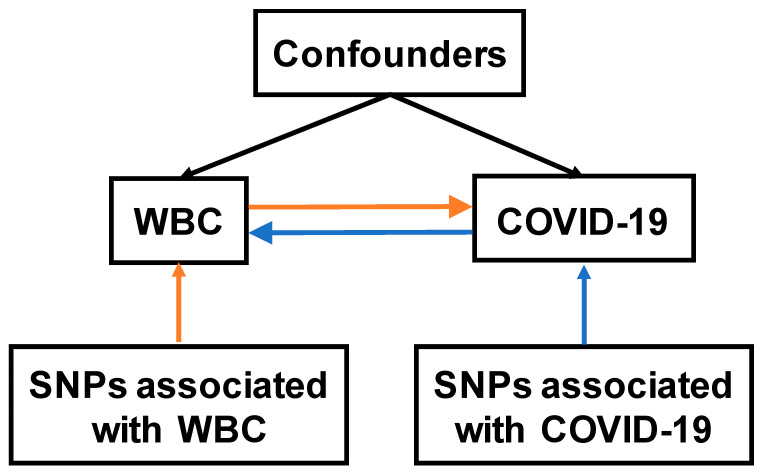
Overview of two-sample MR study on the bidirectional associations between WBC and COVID-19. The forward MR analysis evaluating the causal effects of WBC on COVID-19 was indicated with orange arrows, while the reverse MR analysis evaluating the causal effects of COVID-19 on WBC was indicated with blue arrows. WBC, white blood cells; SNP, single nucleotide polymorphism.

**Figure 2 jpm-11-00195-f002:**
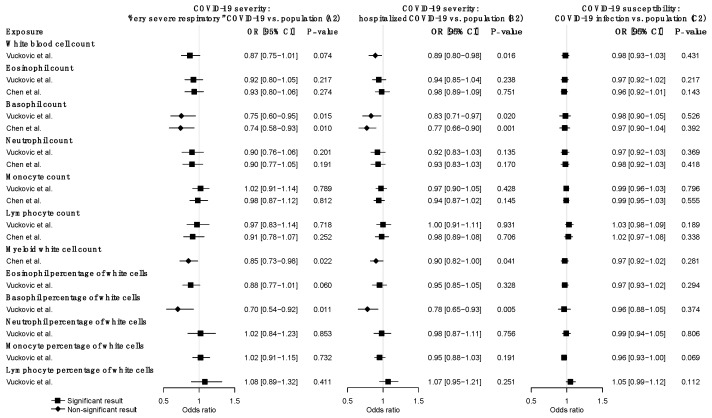
Forward MR estimates of the effects of WBC traits on COVID-19 severity and susceptibility. Odds ratios and 95% confidence intervals were derived using the IVW random-effects model. The *p*-value threshold (0.05) was used for the significant association. Detailed summary statistics could be found in Appendix A.

**Figure 3 jpm-11-00195-f003:**
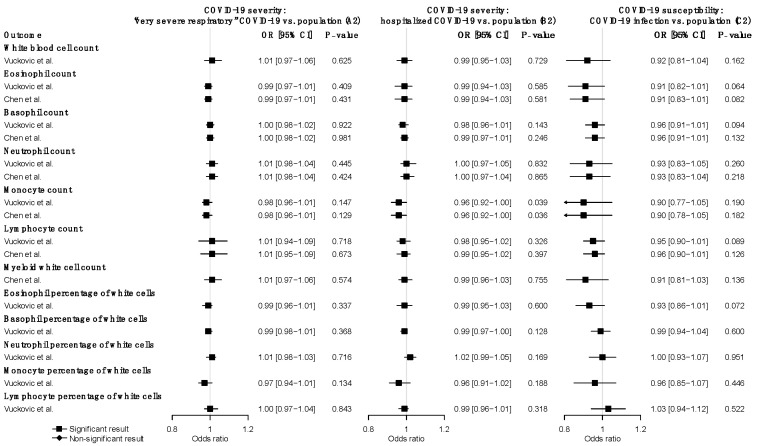
Reverse MR estimates of the effects of COVID-19 severity and susceptibility on WBC traits. Odds ratios and 95% confidence intervals were derived using the IVW random-effects model. The *p*-value threshold (0.05) was used for the significant association. Detailed summary statistics could be found in Appendix A.

## Data Availability

The data are available in Supplementary Files and upon request. Scripts for the MR analysis are freely available on GitHub via https://github.com/yitangsun/WBC_COVID19_MR.

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
