# Peer review of "White Blood Cells and Severe COVID-19: A Mendelian Randomization Study"

_jpm, 2021, doi:10.3390/jpm11030195_

Round 1

Reviewer 1 Report

the manuscript sounds interesting and data provided from authors took several relevant informations regarding covid-19.

the role of basophils in chronic inflammation is well known and authros inderlined it into the text but concerning markers of chronic inflammation , although are not available, much more should be explained: which are comoderators of this link between basophils and inflammation ? proteases? clotting factors, iron metabolism, cytokines? 

i suggest to author improve this section with some sentences in order to give to readers a full explanation of their findings from pathophysiological and clinical points of view.

Author Response

We thank the reviewer for the very positive comments and for providing suggestions for improvement. We added discussions of the potential mechanism between basophils and COVID-19.

“Basophils are known to potentiate humoral immune responses by producing interleukins 4 and 6 (IL-6 and IL-4) [46]. A recent immune-monitoring study of COVID-19 patients found that the relative abundance of basophils increases noticeably from acute to recovery phases and is positively associated with the level of anti-CoV-2 immunoglobulin G (IgG). These observations indicate that the extent of basophil depletion may affect the efficacy of IgG responses to the SARS-CoV-2 infection. The mediating factors between basophils and IgG responses may include IL-6, IL-4, or other explored factors, and the same study also observed a negative association between IL-6 and anti-CoV-2 IgG [47]. The complete elucidation of the potential mechanism warrants further investigation.”

  1. Denzel, A., U. A. Maus, M. Rodriguez Gomez, C. Moll, M. Niedermeier, C. Winter, R. Maus, S. Hollingshead, D. E. Briles, L. A. Kunz-Schughart, et al. "Basophils enhance immunological memory responses." Nat Immunol 9 (2008): 733-42. 10.1038/ni.1621. https://www.ncbi.nlm.nih.gov/pubmed/18516038.
  2. Rodriguez, L., P. T. Pekkarinen, T. Lakshmikanth, Z. Tan, C. R. Consiglio, C. Pou, Y. Chen, C. H. Mugabo, N. A. Nguyen, K. Nowlan, et al. "Systems-level immunomonitoring from acute to recovery phase of severe covid-19." Cell Rep Med 1 (2020): 100078. 10.1016/j.xcrm.2020.100078. https://www.ncbi.nlm.nih.gov/pubmed/32838342.

Reviewer 2 Report

The paper is well written, scientifically sound and of high interest. However, in the light of introduction and discussion I suggest to cite and discuss this paper published on another MDPI journal - DOI: 10.3390/diagnostics10090619

Author Response

Response: We thank the reviewer for the very positive feedback and suggestions. In our revised manuscript, we added this citation to the introduction and discussion sections.

“A growing number of studies have found that immune cells may play important roles in the COVID-19 severity and susceptibility [5-9].”

“Previous systematic reviews and meta-analyses of observational studies have shown WBC traits to be associated with COVID-19 severity and susceptibility [5-7, 9].”

  1. Sambataro, G., M. Giuffre, D. Sambataro, A. Palermo, G. Vignigni, R. Cesareo, N. Crimi, S. E. Torrisi, C. Vancheri, L. Malatino, et al. "The model for early covid-19 recognition (mecor) score: A proof-of-concept for a simple and low-cost tool to recognize a possible viral etiology in community-acquired pneumonia patients during covid-19 outbreak." Diagnostics (Basel) 10 (2020): 10.3390/diagnostics10090619. https://www.ncbi.nlm.nih.gov/pubmed/32825763.